# Wood as Possible Renewable Material for Bone Implants—Literature Review

**DOI:** 10.3390/jfb14050266

**Published:** 2023-05-10

**Authors:** Vadims Nefjodovs, Laura Andze, Martins Andzs, Inese Filipova, Ramunas Tupciauskas, Linda Vecbiskena, Martins Kapickis

**Affiliations:** 1Faculty of Residency, Riga Stradins University, Dzirciema iela 16, LV-1007 Riga, Latvia; 2Microsurgery Centre of Latvia, Brivibas Gatve 410, LV-1024 Riga, Latvia; 3Latvian State Institute of Wood Chemistry, Dzerbenes Street 27, LV-1006 Riga, Latvialinda.vecbiskena@gmail.com (L.V.)

**Keywords:** wood implants, bone repair, biocomposites, osteosynthesis

## Abstract

Bone fractures and bone defects affect millions of people every year. Metal implants for bone fracture fixation and autologous bone for defect reconstruction are used extensively in treatment of these pathologies. Simultaneously, alternative, sustainable, and biocompatible materials are being researched to improve existing practice. Wood as a biomaterial for bone repair has not been considered until the last 50 years. Even nowadays there is not much research on solid wood as a biomaterial in bone implants. A few species of wood have been investigated. Different techniques of wood preparation have been proposed. Simple pre-treatments such as boiling in water or preheating of ash, birch and juniper woods have been used initially. Later researchers have tried using carbonized wood and wood derived cellulose scaffold. Manufacturing implants from carbonized wood and cellulose requires more extensive wood processing—heat above 800 °C and chemicals to extract cellulose. Carbonized wood and cellulose scaffolds can be combined with other materials, such as silicon carbide, hydroxyapatite, and bioactive glass to improve biocompatibility and mechanical durability. Throughout the publications wood implants have provided good biocompatibility and osteoconductivity thanks to wood’s porous structure.

## 1. Introduction

Bone fractures first drew prehistoric humans’ attention up to 46 thousand years ago during the Early Upper Paleolithic age, the period from which the first healed bone fractures were found by archeologists [1]. After thousands of years of using traction and immobilization as the only treatment for bone fractures, the first true external fixation was applied only 120 years ago. That was developed by a Belgian surgeon Albin Lambotte. Lambotte who also introduced the term “osteosynthesis”—fixation of bone by using mechanical devices [2]. At the beginning of the 20th century, with the development of antiseptics, anesthesiology, and bone imaging possibilities, the modern principles of the internal fixation of fractures were developed. The first material for osteosynthesis implants was nickel-coated steel, developed in the 19th century [3,4]. Other metals such as silver [5], aluminium, and brass [6] have been used to produce different bone implants. Nevertheless, these materials were found not to be fully suitable due to inadequate mechanical properties and corrosion. The first successful material was stainless steel, later joined by titanium and cobalt-chromium alloys [7]. Although the problem with obvious and quick corrosion was resolved, there are still a few debatable issues. First, the density of a metal alloys is up to three times higher than cancellous bone [8,9]. Thus, aseptic loosening of the metallic implants is considered a possible complication within 15 years after surgery [10]. Second, bio-corrosion of stainless steel [11] and titanium [12] alloys is being investigated as well. Demand for non-metallic implant materials is growing, not only because of bone damage over time due to the loosening and biocorrosion of implant material, but also because of the increased use of modern medical diagnostic systems, e.g., nuclear magnetic resonance [13]. Metal implants cause significant artifacts in computer tomography and magnetic resonance images. The lower image quality of artifacts cause blurring. In the last decades, numerous studies have been published about reducing the effects of artifacts. However, the issue is still present in everyday clinical practice [14,15].

Aside from bone fractures, bone defects are a common issue in orthopaedic and reconstructive surgery. Bone defects can be caused by severe injuries, congenital anomalies, and tissue resection due to oncological masses. Although bone has great capabilities for rejuvenation, the healing of large defects is challenging. Treatment with bone xenografts (grafts from animals) from dogs and goats for cranioplasty was first described more than 500 years ago by Ottoman empire surgeon Ibrahim Bin Abdullah [16]. Nowadays, bone defect reconstruction still relies mostly on autologous (from the patient), allogeneic (from another human donor), and xenogeneic (animal-derived) bone grafts. For very extensive defects, vascularized bone flaps are harvested from the patient. Harvesting bone tissues from the patient adds additional surgical sites, with possible complications. Using allogeneic and xenogeneic grafts posts immunological challenges, as well as logistical and ethical issues [17]. In attempts to improve bone defect reconstruction, different biomaterials have been widely investigated—calcium phosphates [18,19], bioactive glass [20], collagen [21], silk fibroin [22], etc.—for potential use in clinical practise [23].

Great interest has been observed among industry, researchers, and society towards new materials produced from natural sources, due to several reasons, e.g., topicality of environmental protection issues, new regulations, unsustainability of fossil fuels and their reserve, increasing plastic pollution, and global concepts of circular bio-economy [24]. Reducing emissions in the next decade is crucial for Europe, which intends to become the world’s first climate-neutral continent by 2050 by making the European Green Deal a reality [25]. Concerns about environmental issues have encouraged research on biomaterials, including bone implants [26]. Over the last few years, comparative studies have been carried out on the ecological footprints of raw materials for bone implants [26,27], as well as studies on the use of natural or bio-based polymers in bone tissues [27,28,29], the synthesis of hydroxyapatite from sustainable natural raw materials [30,31], and the use of new technologies such as 3D printing as a solution for a sustainable and circular economy [32]. Materials obtained directly from nature are being studied as well, e.g., corals [27]. However, the materials developed thus far do not provide sufficient mechanical strength for osteosynthesis implants compared with metal implants. These issues promote further investigations for alternative implants’ material.

## 2. Similarities between Wood and Bone

Humanity has known about wood as a biomaterial since the Stone Age, with wood has played a major role in humanity’s greatest achievements—from discovering fire to creating transport. Wood is an anisotropic natural material usually obtained from the trunk of a tree. It can be defined as a heterogeneous composite that consists mainly of natural polymers such as cellulose (40–50%), hemicelluloses (15–25%), and lignin (15–30%) [33,34]. Tree cross-sections can be distinguished into three components—the bark, cambium, and wood parts—xylem. The bark consists of a cork layer on the outside and a phloem layer on the inside. The cambium, located between the bark and the xylem, consists of living cells that form the new xylem and phloem layers. Xylem has two wood parts—sapwood and heartwood. The sapwood consists of dead cells and a small number of living parenchymal cells. It acts as food storage, as a water and nutrients transporter, and as mechanical support for the tree. The heartwood consists entirely of dead cells and provides only a support function for the tree [35].

Similar to wood, bone is also an anisotropic heterogeneous composite material, as it consists of about 60% inorganic material (calcium and phosphate in a form of natural or calcium-deficient hydroxyapatite), 30% organic material (collagen) and 10% water [36,37,38]. Similar to wood, which fulfills the function of support in a tree, the main function of bone is to support the static and movement functions of the body. Bone also acts as storage for minerals such as calcium and phosphate, as well as in the maintenance of homeostasis [39]. Three parts of bone can be observed in the cross-section—cortical bone, also called dense or compact bone, trabecular bone, also called spongy or cancellous bone, and bone marrow cavity. Bone tissue contains three main cell types: osteoblasts, osteoclasts, and osteocytes. Osteoblasts are responsible for bone formation, while osteoclasts are cells that resorb bone. Bone homeostasis is maintained by the connection between bone formation and bone resorption (bone turnover). Osteocytes are cells that are found in fully formed bone and make up most of the bone [40]. Similarities between wood and bone have been observed and were described as early as the invention of the microscopic magnification itself. The pioneer of microscopy, Antonie van Leeuwenhoek, described the analogy between the osteoid bone structure and the fibre structure of wood [41]. Since then, several authors have continued researching similarities between wood and bone, revealing the hierarchical macroscopic and microscopic structures of both, as well as functional similarities such as biomechanical characteristics, remodeling, and liquid transportation abilities [42,43,44,45,46]. Figure 1 shows the structural similarity between cortical bone and wood at the micro and nano level. 

The strength of both cortical bone and wood is ensured by its structural construction. The cortical bone base is formed by osteons, while wood consists of wood cells. Both the osteon and the tree cell are oriented in the direction of the long axis of the bone and wood, respectively, and are composed of several concentric layers of parallel fibers or fibrils. Each layer is oriented in different directions, thus providing mechanical strength. In bone, the layers are formed of collagen fibers, while wood cells are composed of bundles of cellulose micro fibrils. Collagen fiber consists of collagen fibrils constructed from triple helix collagen molecules and mineral nanocrystals, while cellulose microfibril is constructed from amorphous and crystalline parts of parallel cellulose molecules [36,47,48]. Based on similarities between wood and bone, wood has been used as a testing model for orthopedic implants [49]. Despite the structural similarity between wood and bone, solid wood has not been amply considered as a possible biomaterial for bone implants. Only in recent decades has wood been studied by a few authors as a possible implant material.

Considering the mentioned topical problems, this review presents a brief overview of the use of wood as a biomaterial for bone implants, emphasizing the wood processing and current research methodologies. 

## 3. Wood Species for Bone Implants

Individual species of trees have been studied as a source of biomaterial. Most of the published studies have main purpose of creating biomaterials for bone defect substitution and rarely for creating wood-based orthopedic implants.

### 3.1. Birch

One of the first studies regarding wood as a possible implant material was performed by Kristen, Bösch et al. using birch wood [50]. Birch is one of the most widespread and economically important species of deciduous trees in Europe and Scandinavia. Silver birch (*Betula pendula*) and European white birch (*Betula pubescens*) are among the most common birch species found in most of Europe, up to Central Siberia. *Betula pubescens*, which is the northernmost tree species, is more common in the Northern and Eastern parts of Europe. *Betula pendula* is more common in the southern regions of Europe, such as the Iberian Peninsula, southern Italy, and Greece [51]. Birch wood has an average density of 600–650 kg/m^3^ and a high Jank hardness of 4000–5000 N, and contains little extractive material, which makes this wood well suited for bone implants [52,53]. The early studies were conducted in vivo using rabbits. Birch implants were pre-treated with ethanol and placed transcortically into rabbit tibias. Evaluation was done after 3, 5, 14, and 32 weeks. Although the tissues produced a foreign body reaction, a new bone formed around the wood implants. Additionally, bone ingrowth into the implant’s pores was recorded [54]. Similar ethanol pre-treated birch implants were implanted into a rabbit’s soft tissues. After controls within 2, 6, 12, and 30 weeks, it was concluded that ethanol pre-treatment was not sufficient to prevent a foreign body reaction [50]. Years later, Rekola, Aho et al. published a novel wood pre-treatment method—preheating of birch implants at different temperatures—140 °C, 200 °C, and 220 °C for 2 h. The implants were placed into the drilled cavities of rabbit femurs and observed after 4, 8 and 20 weeks detecting the bone ingrowth. Preheated birch implants showed better osteoconductivity compared to untreated implants. However, when applying the highest temperature of 220 °C, the biomechanical characteristics of the implants were decreased [55]. In vitro studies were performed by immersing the birch implants into simulated body fluid (SBF) for 63 days at 37 °C. It was documented that immersion in the SBF significantly decreases the biomechanical properties of the untreated implants, while heat pre-treated implants preserve these properties [56,57].

### 3.2. Ash

Ash is a tree of the olive family that is widespread in Europe, Asia, Canada, and North America. As a hardwood with a low content of extractive substances, a high density of 600–680 kg/m^3^, strength, and flexibility, it is suitable for bone implant materials [58]. The in vivo study with ash implants was conducted simultaneously with early birch studies. Ash specimens were ethanol-pretreated and fixed in rabbit calcaneus bones with Achilles tendons reattached and analyzed after 5 and 14 weeks. The ethanol pre-treatment of ash resulted not only in bone ingrowth, but the tendons’ tissues grew into the wood pores as well, along with moderate foreign body reaction [59].

### 3.3. Lime, Willow, and Fir

Spruce wood is widespread in Scandinavia, Northern Eastern Europe, North America, Canada, and Japan, and is one of the most economically important coniferous wood species [60]. The white willow (*Salix alba*) is the most well-known of the willows, widely distributed throughout Europe except for the most northern regions. The northern part of Europe, where willow is common, includes the British Isles, the Netherlands, and the Baltic coast (Latvia and Lithuania). Willow is also found in Mediterranean regions, as well as in North Africa (Morocco and Algeria) [61]. Lime trees are common in Eastern North America (*Tilia americana*) and Europe (*Tilia Europen*; hybrid wood). All named wood species have a low density—400–450 kg/m^3^ for fir and willow, 450–550 kg/m^3^ for lime wood. These wood species, together with birch and ash, were used to fix fractures in rabbit femurs by Horsky, Huraj, Paukovic. Implants were untreated before implanting in vivo. Birch, ash, and fir were well tolerated, while lime and willow caused acute inflammatory reaction, indicating that differences in wood species meant that not all species would be suitable for bone implants [62]. Although all the mentioned wood species contain a high extractive content, they differ in their composition. Fir extract contains the most lignans [63], while willow extracts contain a large amount of salicylic compounds, flavonoids, and tannins. These substances are bioactive compounds characterized by antipyretic, analgesic, anti-inflammatory, antirheumatic, and anticoagulant properties. As with all bioactive substances, they can be toxic at certain levels [64,65].

### 3.4. Juniper

Juniper is the world’s most widespread and northernmost coniferous tree. It is common both in Europe and Asia, as well as in North America and Japan. Juniper can be found both in the farthest North areas of Scandinavia and in the mountain areas of the warmer regions of Southern Europe. The density of juniper is 450–600 kg/m^3^ [66]. Juniper has long been studied for its antibacterial properties, but not for use in bone implants. The essential oils in juniper wood can also be toxic at high dosages; therefore, pre-treatment is required [67]. A unique in vivo study considered juniper wood for potential orthopedic hardware. Hip prostheses were crafted and pre-treated in boiling water for 10 min. The proximal part of rabbit femurs were resected and hemiarthroplasty with the juniper prostheses was performed. Rabbits were allowed to bear weight with no restrictions. Histological analysis was done after 3, 6, 18, and 36 months. No foreign body reaction was documented in any specimens. Initial bone ingrowth was detected after 6 months. After 3 years, wood implants were fully integrated with bone tissues (Figure 2). Essential oils from juniper were tested for their capacity to induce a toxic response in rats and was demonstrated to be well tolerated, especially when released slowly [68]. Almost 20 years later, preliminary studies have been carried out for the possible development of bone implants from partially delignified and compressed solid juniper wood, thus improving the mechanical properties of the implant. A compressed wood density of 1170 kg/m^3^ was achieved (100% increase compared to natural juniper wood). The modulus of rupture was increased by 85%, reaching 174 MPa, and the modulus of elasticity by 620%, reaching 12,500 MPa [69].

### 3.5. Carbonized Wood

Another trial for the development of bone implants has been proposed by pre-treating wood at high temperatures to create a charcoal-type material. In one of the earliest studies, wood from clematis was carbonized at 850 °C for 5 h. Samples were implanted in vivo into rabbit bone, whose tissue was able to grow into the carbonized wood [70]. A similar in vivo study was performed with bamboo charcoal. The results showed that charcoal bamboo as a bone substitute has good biocompatibility and osteoconductivity [71]. Although pure carbonized wood had good biocompatibility and osteoconductivity, the complete loss of its mechanical properties made it an impractical material. Years later, the mechanical properties of pure carbonized wood were improved by an impregnation with silicon carbide (SiC) to produce a biomaterial called ecoceramics. In the preparation process, natural wood was pyrolyzed at 1000 °C using argon gas; the natural wood lost around 75% of its weight and 60% of its volume as a result of the treatment. The remaining scaffold was infiltrated with melted Si at 1550 °C. Si reacts with carbon in pyrolyzed wood to form SiC. Different wood species have been used to produce wood-based ecoceramics, for example, maple [72], eucalyptus [73], mango [74], oak [75], beech [76], pine [77], and others [78]. The technique preserved the porous structure of the wood while adding the rigidity of SiC. It is also a light-weight material, with density around 1100–2300 kg/cm^3^, depending on the selected wood [74,75,76]. In addition, ecoceramics have great heat and electric resistance [79,80]. Due to various properties, ecoceramics have attracted more interest of researchers in civil [81], aeronautical [82] and electronic [83] engineering, and only a small number of studies consider ecoceramics as a material for medical applications. One in vivo study has been done with SiC scaffolds that were implanted in sheep metatarsal bones. Histological analysis was performed after 4, 8, 12, and 48 weeks. Analysis revealed good scaffold-to-bone adhesion, and new bone ingrowth inside the scaffolds was documented as well [84]. Since then, few authors have proposed combining wooden scaffolds with other biomaterials. In one study, SiC scaffolds derived from beech, eucalyptus, and sapele were combined with bioactive glass. An in vitro study with MG-63 osteoblasts showed good cellular attachment to both coated and uncoated SiC scaffolds. Additionally, the osteoblasts proliferated equally in standardized environments and on the surface of bioactive-glass-coated SiC scaffolds [85]. In another study, carbonized wood scaffolds derived from cane and pine [86,87] or rattan [88] were combined with hydroxyapatite (HA). The obtained samples showed the preserved porous structure and improved mechanical properties; the compressive strength reached 0.4 MPa and the tensile modulus increased 2–3 times [86,87,88].

### 3.6. Cellulose-Based Scaffold

Few authors have considered wood as a base for cellulose-based scaffolds. To create such scaffolds, more extensive wood processing is required. Firstly, wood is processed into cellulose. Cellulose is a natural linear cell polysaccharide consisting of glucose (C_6_H_10_O_5_)_n._ (Figure 3). It is the main component of cell wall in green plants and algae, and bacteria produce cellulose to form a biofilm as well. While the purest natural form of cellulose is cotton, where cellulose comprises about 90% of cotton’s mass, wood is made of around 57% cellulose and remains the main source for producing cellulose [89]. As a natural raw material, cellulose has been used for fabrics and papers for hundreds of years, but only 185 years ago, in 1838, the chemical structure of cellulose was discovered and described by French chemist Anselme Payen [90]. Since then, production of cellulose from wood stock is performed by chemically dissolving unwanted components such as lignin, short-chained polymers, etc. Cellulose is widely used for its porous structure and insolubility in water and organic substances in medical filters [91], pharmacy [92], and wound dressings [93]. In the last few decades, cellulose has also attracted researchers’ attention as a potential biomaterial for medical applications, similar to using wood as an implant material. In the 1960s, implantation of cellulose sponges was used to study tissue inflammation and granulation formation shortly after implantation [94]. Years later, in the 1990s, researchers began to investigate the long-term effects of cellulose implantation. Märtson, Viljanto et al. used industrial soft cellulose sponges derived from eucalyptus, birch, or oak. Cellulose sponges were tested in vivo in rat soft tissue, and histological examinations were performed consecutively after 1–60 weeks. Histological evaluation revealed that the inflammatory response of the surrounding tissues subsided after 4–6 weeks and revealed good connective tissue ingrowth into the cellulose sponges. The researchers also detected a slow resorption and degradation of the pure cellulose sponges [95,96]. In addition, the biocompatibility of cellulose sponges with bone tissue was investigated in vivo; cellulose sponges were tested into the femoral bone cavity of rats. Bone ingrowth into cellulose sponges was recorded after 4–6 weeks [97]. Later researchers started combining cellulose fibers with other biomaterials. An in vitro study was performed with chondrocytes from the bovine knee joint. Cellulose scaffolds were exposed to saturated calcium hydroxide (Ca(OH)_2_) solution, then immersed in supersaturated simulated body fluid (SBF). Thus, a calcium phosphate coating was created. Although the cellulose and calcium phosphate scaffolds caused an acidic reaction in solution and the pH had to be adjusted with calcium hydroxide [98], better cellular adhesion was detected compared to untreated cellulose scaffolds. In another study, cellulose fibers were impregnated with hydroxyapatite particles. Tomilla, Ekholm et al. published two studies on cellulose coating with hydroxyapatite derived from bioactive glass. Bioactive glass S53P4 (Abmin Technologies Ltd., Turku, Finland) was dissolved in SBF, and cellulose sponges were immersed in the SBF solution at 37 °C for 24 h. After 24 h of immersion in SBF, calcium hydroxyapatite was formed on the surface of the scaffold. In an in vivo study with scaffolds implanted in rat soft tissue, an acute inflammatory response was reported on the first day after implantation. More extensive connective tissue formation was observed in the hydroxyapatite layer, while the inflammatory response disappeared within 14 days [99]. In another in vivo study, biomimetically coated cellulose sponges with silica-rich apatite were implanted into femoral bone defects in rats. After 12 weeks post-implantation, apatite-coated cellulose sponges did not significantly improve bone ingrowth compared to uncoated cellulose sponges. [100]. Later, Daugela, Pranskunas et al. investigated cellulose-based scaffolds substituted with micro- and nano-hydroxyapatite particles. An in vitro study was performed on human-like osteoblastic cells (Mg-63) to determine cytotoxicity and cell adhesion. According to the results, cell adhesion was improved by hydroxyapatite nanoparticles compared to cellulose-based scaffolds substituted with hydroxyapatite microparticles, and no cytotoxic response was detected. Similar to the in vitro results, scaffolds with hydroxyapatite nanoparticles significantly improved bone tissue ingrowth in rabbit calvaria bones [101]. Further studies also involved cellulose derivatives; carboxymethyl cellulose scaffolds were prepared using a freeze-drying process. An in vitro study showed that prepared scaffolds supported the proliferation and differentiation of Saos-2 cells, and extensive tissue proliferation was detected in rat subcutaneous tissues in vivo [102]. Cellulose scaffolds have also attracted interested as a localized drug-delivery system in damaged bone. Different studies have been published on the delivery of growth factors, bioactive proteins, antibiotics, and anti-inflammatory drugs [103].

Table 1 summarizes brief facts about the wood species studied in the literature.

## 4. Discussion

### 4.1. Advantages and Disadvantages of Wood as Bone Implants

The most important advantage of wood as a bone implant is the structural similarities between bone and wood, described by the pioneer of microscopy, Anton van Leeuwenhoek, back in 1693. Since then, the structural and functional similarities have been described by many other authors [41,42,43,44,104]. This was discussed in the previous sections.

Wood is a natural composite material consisting of three main components—cellulose, hemicelluloses and lignin [105]. Simply, each of the components gives the wood specific mechanical properties—strength, flexibility, and stiffness, respectively. The chemical composition of wood is variable and depends on the wood species, age, genetic factors, and growing conditions. [106,107,108] In principle, this could be considered a disadvantage, but this property of wood as a biomaterial is also an advantage. Knowledge of these effects on wood properties can help in finding suitable wood materials for a specific application. Furthermore, chemical [109,110], thermal [111,112], or enzymatic [113,114] treatment can change the chemical composition of wood, thus affecting its mechanical properties. 

Porosity is another advantage of wood as a biomaterial. Pore size distribution in the wood varies from 1 nm to 100 µm and can be classified into macro-, meso- and micropores. Porosity is inversely proportional to wood density [110,115]. Osteoblasts are approximately 10–50 μm in size and require 100–200 nm pores for ingrowth and bone regeneration. In the pores with a smaller size, the formation of osteoid and fibrous tissue occurs [116]. It can be concluded that the ingrowth of bone cells will occur more easily in less dense wood. At the same time, wood density directly affects the mechanical properties of wood. The porous structure of wood easily lends to impregnation and can be used to introduce bioactive substances or drugs for bone regeneration.

The biodegradation of wood is one of the possible reasons why wood as an implant material causes skeptical reaction. To justify this property of wood, it is important to consider that wood decay occurs at a certain humidity and in the presence of oxygen. In a highly wet and oxygen-free environment, wood biodegradation occurs very slowly and wood can be preserved for hundreds of years [117,118]. Water content in human body lean mass (or fat-free mass) is around 70–75%; thus, it is considered a non-oxygen wet environment when inserting a wood implant [119]. 

Shrinking and swelling in water is a typical characteristic of wood. The amount of water in the wood is significantly affected by the humidity and temperature of the environment. Accordingly, in a dry environment, the wood dries quickly and cracks form; conversely, in a water environment, the wood absorbs moisture and swells. This characteristic complicates the use of wood materials and requires evaluation of the preparation process of wood samples. The water content of natural green wood is approximately 60–70%. The moisture content of dry wood is approximately 7%. Considering that the water content in human muscle mass is 70–75%, it is recommended to keep the moisture content in wood above 50% when developing a wood implant, thus preventing possible problems that may arise due to shrinking and swelling. It should be noted that the water content in solid wood affects its mechanical properties. The elasticity of wood is directly proportional, while the strength is inversely proportional to the water content of the wood [120,121].

Other disadvantages of wood as a biomaterial for bone substitution are its uneven and different properties, variable density and chemical composition depending on the wood species, growing conditions, genetic aspects, and age [105,107,108]. These makes it difficult to obtain materials with the same properties. One of the solutions is to use wood from plantations or to obtain pre-treated wood. 

Apart from the basic components of wood—cellulose, hemicellulose, and lignin—it also contains a small number of inorganic compounds (up to 1%) and soluble organic compounds called extractives. The presence of extractives in wood samples can be critical for their use in bone implants. The extractives consist of mixtures of various components, from relatively low-molar-mass molecules to the higher molar-mass substances [122] such as fats, fatty acids, waxes, sterols, terpenic compounds, phenolic compounds, pectins, flavonoids, stilbenes, tannins, etc. [120] Some of the compounds are bioactive substances and, depending on the concentration, can be either therapeutic or toxic to the human body. Extractives can be divided into groups based on their chemical type—lipophilic or non-polar and hydrophilic or polar compounds. Each of the mentioned groups can be dissolved in different solvents—organic solvents or water. To eliminate all extractives, consecutive extraction is performed using different solvents of increasing polarity, e.g., dichloromethane, acetone, ethanol, and water [123]. Since the human body comprises 70–75% water, separation of the water-soluble extractives from the wood before implantation is critical. Pre-treatment of wood with both ethanol and water is necessary so that the extractives do not cause toxic reactions.

As a biological material, wood also provides a habitat for various microorganisms, such as fungi and bacteria, that are not desirable in bone implant material. Considering the above, special attention should be paid to the chosen sterilization methods. Not all popular bone implant sterilization methods are applicable to wood samples. Such classical methods as UV, ethanol, or ethylene oxide treatment [124], autoclaving, or steam treatment [125] cannot be used for wood materials. Wood is destroyed under the influence of UV [126], swelling occurs during water vapor treatment, and wood hydrolysis begins at elevated temperatures above 140 degrees [111]; but in the case of ethanol or ethylene oxide (toxic) treatment [124], it could be problematic to ensure the removal of all substances from the sample due to its porous structure. It is possible to use gamma irradiation or microwave treatment [127], but in this case, a suitable processing time should be chosen, as the wood may be destroyed due to heating (more than 140 degree) [128,129]. Sterilization with gaseous phase compounds such as supercritical CO_2_, hydroxyl peroxide, or peracetic acid is applicable to porous fibrous materials, including wood [124,130,131].

Additionally, greater mechanical properties and density are needed to use wood in osteosynthesis implants. Densification of wood increases the mechanical properties and density of wood. Chemical pretreatment makes it possible to reduce the variability of the chemical composition. Chemical pretreatment of wood and subsequent densification is a promising method for wood processing to obtain implants with density and mechanical strength suitable for osteosynthesis biomaterials [69].

### 4.2. Mechanical Properties of Wood as Bone Implant Compared with Other Implant Materials

Natural wood has a density from 450 up to 700 kg/m^3^, depending on species. The higher density is observed in hardwood species, whereas softwoods have lower densities. The modulus of elasticity (MOE) reaches 1550–13,500 MPa and the modulus of rupture (MOR) reaches 60–100 MPa [132,133]. Mechanical properties can be altered by various methods such as applying heat and chemical treatment. Applying lower heat, up to 200 °C, can increase the MOE, and rupture can be increased by up to 50% [134]. If wood is heated over 800 °C, it loses up to 80% of its mass and 60% of its volume. The obtained carbonized wood has poor mechanical properties, with a density around 200–400 kg/m^3^, although biocompatibility and osteoconductivity is preserved [71,135,136]. Another technique for wood processing is densification, which involves partial delignification and compression. Authors Andze L. et al., in a mechanical study, increased juniper’s density by 100%, reaching almost 1200 kg/m^3^. The MOR and MOE were increased by 85% and 620%, accordingly [69]. Studies of wood’s mechanical properties show that natural wood is not strong enough to produce durable orthopedic implants for bone fracture fixation. Nevertheless, with certain processing mechanical properties of wood can be improved to fit requirements for orthopedic implants. The most common material for orthopedic implants is still titanium and its alloys. Titanium has a density of 4500 kg/m^3^, MOR of 45,000 MPa and MOE of 120,000 MPa [137]. Obviously, titanium’s mechanical strength is multiple times higher than any biological material, including bone—its density is up to 1200 kg/m^3^ [138,139,140], its MOE varies from 10–3000 MPa, and its MOR is 150–180 MPa. [141,142] This significant disparity in mechanical properties allows stable fixation for fractures, but can also cause complications such as aseptic loosening [143,144,145,146,147,148].

Since other materials used in bone repair are dedicated to bone defect substitution, their mechanical properties are unessential, as they are not supposed to provide mechanical support to the bone [149]. Among the investigated biomaterials are tricalcium phosphate bioceramics (density 3070 kg/m^3^, MOR 1.3 MPa, MOE 49 MPa [150]), hydroxyapatite bioceramics (density 3050 kg/m^3^, MOR 18 MPa, MOE 174 MPa [151]) bioactive glass 45S5 (density 2850 kg/m^3^, MOR 45 MPa, MOE 60 MPa [152]), collagen (density 2700 kg/m^3^, MOR 2 MPa, MOE 46 MPa [153]), and silk fibrion (density 1400 kg/m^3^, MOR 5 MPa, MOE 100 MPa [154]). Mechanical properties of different materials are summarized in Table 2.

### 4.3. Summary for Further Investigation

The interest in biomaterials has been constantly growing in the past decades. As environmental issues are a growing concern, an alternative to unsustainable and non-renewable materials is being developed [24]. This direction of development also includes medical implants. Although metallic implants have been greatly improved in terms of biocompatibility and corrosion resistance, their environmental impact is impossible to avoid [155]. Multiple studies have proven that sustainable biomaterials are not only suitable for the production of different implants, but their production process also has a significantly lower ecological footprint compared to any non-sustainable resources [27,29,156]. In the search for suitable biomaterials to be used in bone repair, wood has been one of the potential options studied. The earliest studies published in the last century were conducted using in vivo models. These studies proved the osteoconductive abilities of natural wood implants. Osteoconductivity is a passive attribute of implants, where they allow bone tissue ingrowth on the surface or inside pores of an implant [157,158,159]. This feature of wood implants has been proven in a few studies by obtaining microscopic pictures of new bone trabeculas inside wood pores [54,55,68]. Another important feature of bone implants is osseointegration—direct contact and anchorage between bone and implant, which is maintained over the long-term [160,161]. This feature was demonstrated with juniper prosthesis in rabbits. Animals were able to bear weight with no restrictions up to 3 years, without any implant failures [68]. Studies done in the 21st century are executed in vitro. Throughout the studies, not every species of wood had proved to be equally suitable for bone implants; e.g., lime and willow showed an acute inflammatory reaction [62]. It was concluded that the inflammatory reaction depends on the soluble substances of wood (extractives). Additionally, bamboo before treatment produced cytotoxicity in an in vitro study [162]. Other species such as birch, ash, and juniper have presented excellent biocompatibility [54,55,59,68]. To reduce a possible inflammatory reaction, pre-treatment of the wood is a crucial step. Not only for the sake of asepsis; additional components in untreated wood, e.g., fungi, have been found [163]. They alone can produce an inflammatory reaction [164]. Techniques for pre-treatment differ in various articles. Juniper for an in vivo study has been pre-treated only in boiling water for 10 min [68]. Other authors have tried ethanol pre-treatment that was not sufficient to avoid an acute inflammatory reaction [59]. In the following studies, pre-treatment with higher temperatures became more popular. Aho, Rekola et al. used heat as high as 220 °C for birch and ash implants [55]. This range of temperature neutralizes bacteria, fungi, and all organic extracts, such as essential oils. Thus, pre-treatment at high temperatures reduces the risk of toxic reaction to the minimum. Other authors considered wood as a scaffold for creating new biomaterials. In the oldest studies, charcoal was researched as a possible biomaterial. Although osteoconductive properties were preserved, poor mechanical properties limited further applications. Since the first articles on wood-derived ecoceramics were published, this biomaterial has gained considerable research interest, as ecoceramics combine wood’s favorable properties such as porosity, mechanical and heat resistance. Aside from bone implant development, these characteristics raise interest in a wide range of industrial uses, i.e., filters, catalysts, electric sensors, etc. [165]. Despite the fact that only afew studies were published on biocompatibility of ecoceramics [166,167], the concept was advanced by combining ecoceramics with other biomaterials, such as HA and bioactive glass. Both concepts of wood as biomaterial—hybrid biocomposites and pure wood—have been highlighted in research as a potential biomaterial for repairing damaged bone [18,19,20,21,22,25,26,27,28,29,30,31,32,33]. It should be noted that other biomaterials are also available for this purpose, such as calcium phosphate bioceramics, bioactive glasses [168], and different composite materials combining bioactive inorganic materials with biodegradable polymers [169,170]. Wood is also a main source for cellulose, which has a wide application for medical devices and wound dressings as well. Although only a small number of studies involving cellulose scaffolds have been devoted to bone surgery, some promising results have been reported for bone tissue proliferation and local drug delivery to improve bone healing [171,172]. Orthopedic implants have demanding mechanical requirements to sustain long periods of mechanical loading. For this reason, thus far, metal implants are dominant, and only two groups of authors have processed wood with a goal of orthopedic implants for fracture fixation and joint arthroplasty [62,68,173].

## 5. Conclusions

Wood is a sustainable and renewable source suitable for the production of biomaterials. The processing of wood is more environmentally friendly, especially compared to titanium production, which emits carbon monoxide and other toxic by-products. Nowadays, there is still limited research on the use of wood in bone implants, despite the fact that its great potential has been demonstrated by available studies. The in vivo studies done in the 20th century’s last decades show great insight into some species of wood’s great biocompatibility and osteoconductivity. Based on the provided review, the continued development of wood implants for further incorporation in surgical practice is suggested.

## Figures and Tables

**Figure 1 jfb-14-00266-f001:**
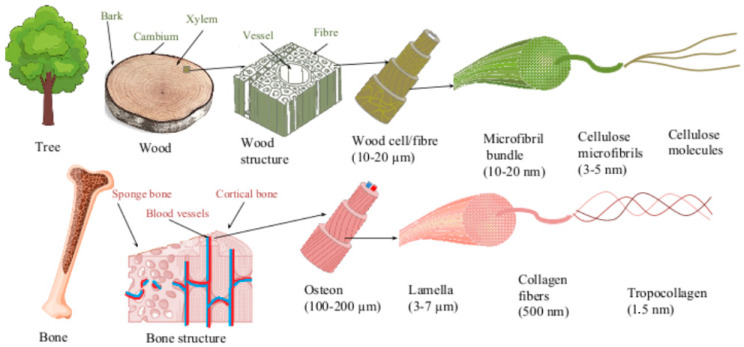
Schematical structure of bone and wood in macro, micro and nano scale.

**Figure 2 jfb-14-00266-f002:**
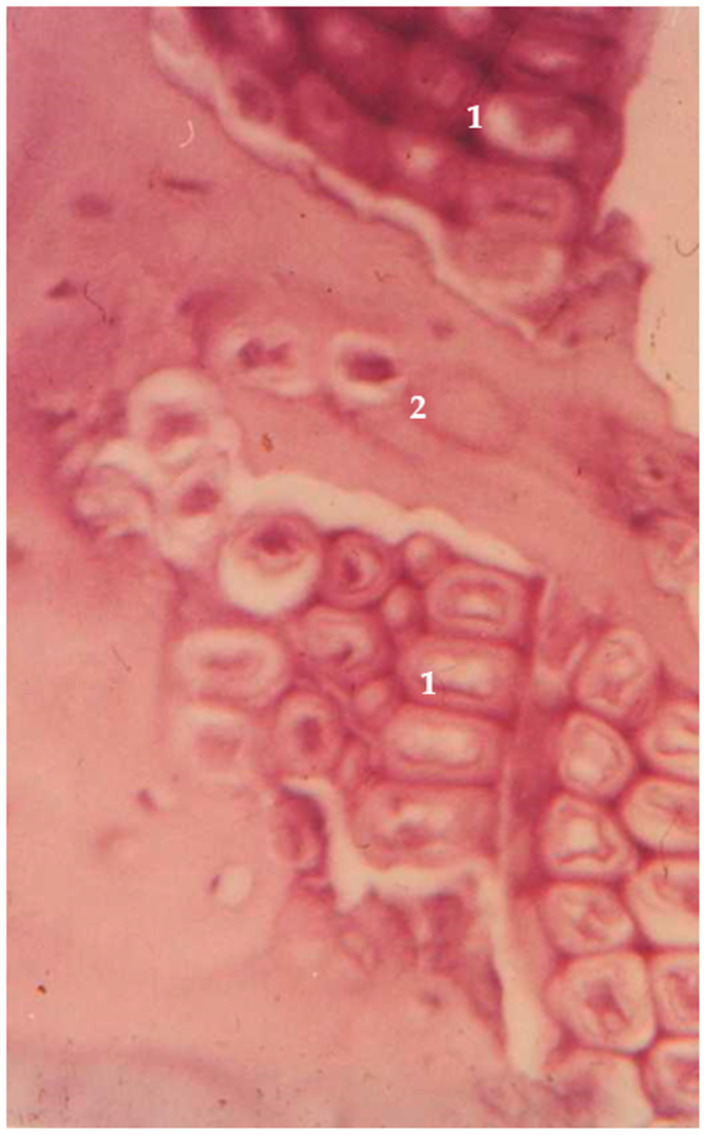
Juniper implant in the in vivo model. 1—juniper implant; 2—bone tissue ingrowth.

**Figure 3 jfb-14-00266-f003:**
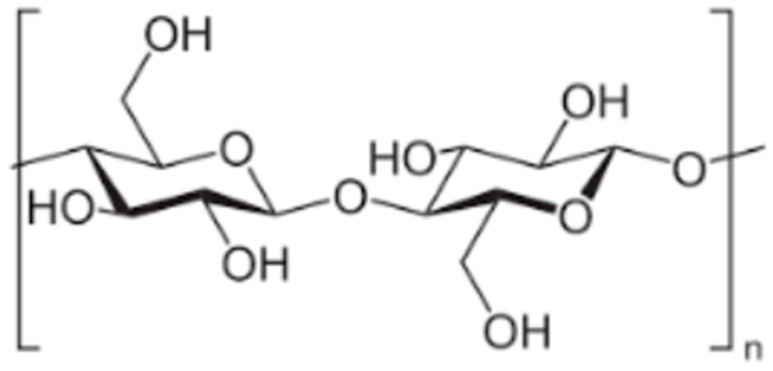
Cellulose molecule.

**Table 1 jfb-14-00266-t001:** Summary of potential wood species for bone implants.

References	Wood Species	Study Design	Results
Kristen, Bosch et al., 1977 [54]	Birch	In vivo	Foreign body reaction after ethanol pre-treatment
Aho, Rekola et al., 2007 [55]		In vivo	After heat pre-treatment good biocompatibility and osteoconductivity
Bosch, Kristen et al., 1979 [59]	Ash	In vivo	After ethanol pre-treatment good biocompatibility and osteoconductivity
Horsky, Huraj et al., 1987 [62]	Fir	In vivo	With no pre-treatment good biocompatibility
	Lime	In vivo	With no pre-treatment acute foreign body reaction
	Willow	In vivo	With no pre-treatment acute foreign body reaction
Gross and Ezerietis, 2003 [68]	Juniper	In vivo	After heat pre-treatment good biocompatibility and osteoconductivity
Andze, Andzs et al., 2022 [69]		Mechanical studies	Partial delignification of wood and subsequent densification showed improved mechanical properties comparable to bone
Kosuwon, Laupattarakasem et al., 1994 [71]	Bamboo	In vivo	Carbonized charcoal showed good biocompatibility and osteoconductivity
de Carlos, Borrajo et al., 2006 [85]	Beech	In vitro	SiC scaffolds allowed good cell proliferation
	Sapele		
Qian, Kang et al., 2008 [86] Tampieri, Sprio et al., 2009 [87]	Cane	Mechanical studies	SiC scaffolds combined with other biomaterials provided significantly improved mechanical resistance
	Pine		
Finardi and Sprio, 2012 [88]	Rattan		

**Table 2 jfb-14-00266-t002:** Summary of mechanical properties for different materials.

Material	Density, kg/m^3^	MOE, MPa	MOR, MPa
Human bone [140,141]	up to 1200	10–3000	150–180
Natural wood [132]	450–700	1550–13,500	60–100
Carbonized wood [135]	200–400	15–140	11–53
Densified wood [69]	1170	12,500	174
Titanium [137]	4500	120,000	45,000
Calcium phosphate bioceramics [150]	3070	49	1.3
Hydroxyapatite bioceramics [151]	3050	174	18
Bioactive glass 45S5 [152]	2850	60	45
Collagen [153]	2700	46	2
Silk fibrion [154]	1400	100	5

## Data Availability

Not applicable.

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
