# Peer review of "Wood as Possible Renewable Material for Bone Implants—Literature Review"

_jfb, 2023, doi:10.3390/jfb14050266_

Round 1

Reviewer 1 Report

The review by Nefjodovs et al. summarises the published literature regarding the suitability of wood for the development of bone implants. Relevant examples are provided for various wood species and wood derivatives. The subject is aligned with the aims and scope of the journal, and might be interesting to a wide audience on account of the increasing need for new biomaterials, especially for bone implants.

Nonetheless, the paper has serious flaws and needs major revision. The following key points should be addressed.

1.     Abstract, “It requires more extensive wood processing, it also possible to combine wood with other materials, like silicon carbide, hydroxyapatite, and bioactive glass”. This sentence is not clear. Please, rephrase (What requires extensive wood processing, and what is it being compared to?)

2.     Introduction, “…the density of a metal alloys’ is up to 3 times higher than cancellous bone [8, 9]. Thus, aseptic loosening of the metallic implants is considered possible complication within 15 years after a surgery”. The density mismatch is certainly a key factor that should be taken into consideration when choosing an implant material. However, aseptic loosening is the outcome of several synergist factors. Tough dating back to almost 20 years ago, a survey of various theories in this regard can be found in the contribution by Sundfeldt et al., https://doi.org/10.1080/17453670610045902

3.     Introduction, “Demand for non-metallic implant materials is growing… also because of increased use of modern medical diagnostic systems, e.g., nuclear magnetic resonance”. This is an interesting point. Could the Authors elaborate on it?

4.     Introduction, “In attempts to improve bone defect reconstruction… for potential use in clinical practise”. A review paper should be as informative as possible. Please, provide additional references, ideally for each class of material.

5.     Introduction, “Wood is an anisotropic natural material… The heartwood consists entirely of dead cells and provides only a support function for the tree [28].” Please add one or more figures that illustrate the multi-layered structure of wood.

6.     Introduction, “…calcium phosphate and calcium hydroxyapatite…”. Why “calcium” hydroxyapatite? What is the difference between “calcium phosphate” and “hydroxyapatite”? The Authors may consider the book chapter by LeGeros and LeGeros, https://doi.org/10.1533/9781845694227.2.367, but this is just an example, as there are numberless references in the literature.

7.     Introduction, “…bone is also an anisotropic heterogeneous composite material… Osteocytes are cells that are found in fully formed bone and make up most of the bone.” Please add one or more figures that illustrate the different types of bone tissue.

8.     Introduction, Figure 1. This figure is key for the paper, as it illustrate the rationale for using wood in bone implants. Accordingly, the quality of this figure should be improved and, ideally, this figure should also be completed with detailed representations of the structure of wood and bone as discussed in the previous comments.

9.     Introduction. The statement “Both the osteon and the tree cell are oriented in the direction of the long axis of the bone and wood respectively and are composed of several concentric layers of parallel fibers or fibrils” and the following statement “Each layer is placed in different directions thus providing mechanical strength” sound mutually contradictory. Please, clarify. A figure would be useful in this regard.

10.  Introduction, “Despite the structural similarity between wood and bone, solid wood has not been amply considered as a possible biomaterial for bone implants.” Could the Authors discuss the reasons why solid wood has rarely been considered for bone implants?

11.  General remark regarding the Introduction: The paper is presently unbalanced, with the Introduction accounting for nearly 30-40% of the main text. In order to remediate, the Introduction should be broken down to shorter sections, for example (this is just a possibility – the Authors are free to consider alternative options):

1)    Introduction

2)    Structural and functional similarity between bone and wood

3)    Advantages and disadvantages of wood as bone substitute (please, note that limitations of wood are also mentioned in the Discussion, and this may cause some overlap - may need restructuring).

12.  Birch, “One of the earliest studies regarding wood as a possible implant material were done by Austrian authors using birch wood”. Please name the Authors of these studies and add the corresponding reference.

13.  Birch, “Birch is one of the most widespread…”: Please, add a world map showing the growth area of birch (and different varieties)

14.  Birch, “… contains little extractive material…”: This is an important point. Please, clarify (what is it? Why is it important when it comes to bone implants?)

15.  Birch, “Birch implants were pre-treated with ethanol…”. Why? With what consequences? What is the effect of the ethanol treatment on wood? Please, expand on this.

16.  Birch, “Years later, birch wood came to the attention of Finish researchers again”. Please, name the Authors and add the corresponding reference. There is no need to specify the nationality.

17.  Birch, “The novelty of the studies was the preheating of birch implants at different temperature…”. As before, why? With what consequences? What is the effect of the heat treatment on wood? Please, expand on this.

18.  Ash, “Ash is a tree of the olive family… and North America”. As already suggested for birch, please add a world map showing the growth area. Please, do the same for all tree species discussed in the paper.

19.  Lime, willow, and fir, “These wood species together…Czech authors”. Please, name the Authors of this research, instead of reporting on their nationality, and add the corresponding reference.

20.  Lime, willow, and fir, “…indicating to differences of wood species not all being suitable for bone implants… can also be toxic at certain levels”. This is a crucial point. Could the Authors elaborate more on this?

21.  Carbonised wood, “the mechanical properties… were improved by an impregnation with silicon carbide (SiC)” and “The remaining scaffold is infiltrated with melted Si at 1550°C.” Is the impregnation with silicon carbide or with melted silicon? Please, clarify.

22.  Carbonised wood, “Various wood species have been used by different authors to produce wood based eco-ceramics - pine, eucalyptus, mango, oak, beech, maple, and others”. Please, provide specific references.

23.  Carbonised wood, “It is also a light-weight material.” What density?

24.  Carbonised wood, “Because of these various characteristics eco-ceramics have interested researchers in civil, aeronautical and electronic engineering more extensively”. Please, provide specific references.

25.  Carbonised wood, “As in one of the study’s authors combined SiC scaffolds derived from beech, eucalyptus and sapele with bioactive glass. … and in the glass coated SiC scaffolds”. This paragraph is not clear. Please, rephrase.

26.  Carbonised wood, “Other authors have combined scaffolds derived from cane and pine [68] [69] or rattan [70] with hydroxyapatite (HA).” Were the scaffolds made of carbonised wood, or eco-ceramics? Please, provide additional detail.

27.  Cellulose-based scaffolds, “Cellulose is a natural linear cell poly- 243 saccharide consisting of glucose (C6H10O5)n.” Please, add a figure that illustrate the molecular structure of cellulose.

28.  Cellulose-based scaffolds, “Cellulose is widely used for its porous structure… in medical filters, in pharmacy and wound dressings”. Please, add specific references.

29.  Cellulose-based scaffolds, “In the earliest publications Finish researchers used industrial soft cellulose sponges derived from eucalyptus, birch, or oak”. Please, name the Authors of this research, instead of reporting on their nationality, and add the corresponding reference.

30.  Cellulose-based scaffolds, “The cellulose scaffold was treated with calcium phosphate”. What kind of treatment?

31.  Cellulose-based scaffolds, “In another studies cellulose fibers were impregnated with hydroxyapatite particles.” Please, add references.

32.  Cellulose-based scaffolds, “Two studies are published on cellulose coating with hydroxyapatite derived from bioactive glass”. What is this?

33.  Cellulose-based scaffolds, “Two studies are published on cellulose coating with hydroxyapatite derived from bioactive glass”. Please, add references.

34.  Cellulose-based scaffolds, “cellulose sponges coated with bioactive glass derived hydroxyapatite”. As before, what do the Authors mean by “bioactive glass derived hydroxyapatite”?

35.  Cellulose-based scaffolds, “another group of researchers investigated cellulose sponge impregnation with micro and nano hydroxyapatite particles.” Please, add references.

36.  Cellulose-based scaffolds, “Further studies also involve cellulose derivates.” Please, add references.

37.  Cellulose-based scaffolds, “One study was performed with carboxymethyl cellulose produced by a freeze-drying process.” Please, add references.

38.  Cellulose-based scaffolds, Table 1. The references in the first column of Table 1 should be numbered as they appear in the References at the end of the manuscript.

39.  General remark regarding Section 2: It has been said that wood can be used according to two strategies, namely “creating biomaterials for bone defect substitution and creating wood-based orthopedic implants” (lines 135-136). Please, rearrange and group the sub-sections of Section 2 correspondingly.

40.  Discussion, “Similarities between bone and wood… have been described by many other authors”. All this has already been said in previous sections.

41.  Discussion. Please, pay attention to terminology. For example, “sterility” is mainly associate with the inability to produce children or young.

42.  Discussion, “the concept was advanced by combining eco-ceramics with other biomaterials, like SiC…”. It was said before (Section 2.5) that eco-ceramics are produced by the impregnation of pure carbonised wood with SiC. How can eco-ceramics be further combined with SiC?

43.  General remark regarding Section 3: at present, Section 3 is just a summary that reiterates (in a short way) what has been written in previous section. It lacks depth and criticality, and fails to reply some basic questions, like:

1)    What are the mechanisms that underpin the bone-bonding ability of wood and its derivatives?

2)    As long as there other biomaterials are also available for substitution of bone defects, such as bioactive glass, bioceramics, hydroxyapatite and other calcium phosphates, and also metals and alloys, what are the advantages and disadvantages of each?

3)    Are different wood species or wood derivates used for different applications (for example, bone defect fillers vs. joint arthroplasty devices)? And why?

44.  Discussion. The limitations of wood as a biomaterial for bone implants should be articulated more clearly and in greater detail, including durability issues, variability, inadequate mechanical properties, presence of contaminant and alien species (fungi, bacteria, even bugs), technical challenges related to sterilisation.

45.  Discussion. Another relevant point to consider is: How can wood and it derivatives be machined to the right shape and size, especially if bespoke devices or complicated geometries are needed?

46.  Conclusions, “Nowadays there is still few research on the use of wood in bone implants despite there was demonstrated its great potential by a couple of studies”. “A couple of studies” sounds reductive, and a couple of contributions cannot really “demonstrate” anything. Please, reword.

47.  Conclusions, “The in vivo studies done in the 20th century’s last decades show great insight into wood’s great biocompatibility and osteoconductivity.” This statement is questionable, as long as many wood species have triggered inflammatory or cytotoxic reactions. Please, rewrite.

48.  English style and grammar should be revised extensively.

Author Response

Dear reviewer,
Thank you for Your valuable recommendations for our article. Sequentially we have answered Your propositions:

  1. Corrections have been made.

  1. Although aseptic loosening indeed is a multifactorial process, the particular study You mentioned focuses on 2 major arthroplasties – knee and hip replacement. Prostheses are fixed to bone with bone cement, besides metal components articulating surface is made of polyethylene or ceramics. Whereas implants for fracture osteosynthesis – plates and screws, are fixed to the bone by titanium screws. Thus, the density mismatch is a much more significant factor in fracture implants compared with arthroplasty implants.

  1. Additional content has been added.

  1. References have been added.

  1. Figure 1. has been improved.

  1. Calcium hydroxyapatite naturally forms in the bones as a form of calcium phosphate (doi: 10.1186/s40824-018-0149-3). The text has been modified to avoid confusion.

  1. Figure 1 has been improved.

  1. Figure 1 has been improved.

  1. Figure 1 has been improved.

  1. This matter is discussed in the Discussion section.

  1. Thank you for the suggestion. We have restructured the introduction.

  1. Corrections have been made.

  1. Since the publication is about the use of wood in bone implants, and not about the distribution and growing conditions of trees, we do not think the inclusion of a world map is necessary.

  1. The information included in the section Advantages and disadvantages of wood as bone implants

  1. We believe the authors chose ethanol treatment as a method of surface sterilization, which was a popular method at the time. Wood itself is inert to ethanol.

  1. Corrections have been made.

  1. Effects corresponded by the authors are reported in lines 163-166.

  1. Since the publication is about the use of wood in bone implants, and not about the distribution and growing conditions of trees, we do not think the inclusion of a world map is necessary. 

  1. Corrections have been made.

  1. One of the authors we cited - Gross K.A. and Ezerietis E. (DOI:10.1002/jbm.a.10437) have tested lethal dosage (LD50) of organic oils of Juniper Communis wood on rats. Results showed that oil was well tolerated, when given orally. high dosing for a long time was necessary to cause lethal result in rats. Similar studies have proved that only high concentrations of essential oil cause significant toxic reactions (e.g. https://silo.tips/download/toxicology-of-essential-oils-reviewed#) however, this information was relevant, when scientists in the 20th century did not use any wood pre-treatment (like heat) before implanting it in-vivo. Studies performed in the 21st century rely on proper sterilization techniques to prepare wood implants. Essential oils components (e.g. alpha-penine found in oil from juniper) have boiling point around 160°C (https://gestis-database.dguv.de/), thus during implant preparation all organic substances are vaporized and deactivated. This point is discussed in lines 326-339

  1. Corrections have been made.

  1. References have been added

  1. References on density have been added

  1. References have been added

  1. Corrections have been made.

  1. Corrections have been made.

  1. Figure 3 has been added.

  1. References have been added.

  1. Corrections have been made, corresponding references are [93] and [94].

  1. More detailed description of the scaffolds’ treatment has been added.

  1. Corresponding references are [97] and [98].

  1. More detailed description has been added (lines 285-288).

  1. Corresponding references are [97] and [98].

  1. More detailed description has been added (lines 285-288).

  1. Corrections have been made; the corresponding reference is [99].

  1. Corresponding references are [99] and [100]

  1. The corresponding reference is [100].

  1. Corrections have been made.

  1. Only one study has published by Gross K.A. and Ezerietis E. (DOI:10.1002/jbm.a.10437) had goal to create an orthopaedic implant (hip prostheses). All the other studies are regarding bone substation. Rather than rearranging the section. The introduction of the section (lines 136-138) has been modified.

  1. Corrections have been made.

  1. Corrections have been made.

  1. Corrections have been made.

  1. 1)    Wood implants have presented osteoconduction and osseointegration in in vivo models (discussed in section 3). The exact mechanisms and reasons why some materials are osteoconductive (like titanium) others are not (like copper) are still unknown (DOI 10.1007/ s005860100282)

2)    As briefly discussed in the paper, non-metallic materials (like bioactive glass, bioceramics, hydroxyapatite and other calcium phosphates) on their own are not dense and strong enough to withhold demanded weight bearing. On the other hand, metallic implants are too dense compared with a bone, which causes mentioned complications like aseptic loosening.

3)    Among described species of wood only juniper has been used for creating arthroplasty device. We believe main favourable factor is juniper’s higher density, compared with other wood species. Therefore, other species (birch, ash, bamboo etc.) have been used only for bone defect substitution, as they are not supposed to provide mechanical strength for the bone.

  1. Thank you for your suggestion – one major section about advantages and disadvantages of wood as bone implants have been written.

  1. There is a lot of possible methods to produce wood materials in specific size and forms, even for small details. For example - laser cutting, mini CNC, waterjet cutting. According to the authors, it is not necessary to include this information in the manuscript.

  1. Corrections have been made.

  1. Corrections have been made.

  1. The manuscript has been revised by a linguistics specialist

Reviewer 2 Report

Dear Editor,

I founded reviewed paper very interesting. It focuses on the promising idea to use naturally occurring, wood-derived components to manufacture medical implants. Authors collected a huge amount of literature to illustrate this idea. Nevertheless, as a clinician I would rather prefer a little bit more information on mechanical and biological properties of described substances and composites produced from them instead of extensive description of history of biomaterials and circumstances of their introduction into the medical practice.

Shortly, I believe that addition of data of mechanical properties of wooden-born materials composed with, at least, cellulose and lignin, and silicon carbide, hydroxyapatite (i.e. Young’s modulus, bending modulus and stiffness) and some facts showing that they are bioinert and biodegradable (together with the dynamics of degradation) would enable the clinician to estimate the usefulness those products as a potential candidates for medical / orthopedic applications. Potentially, these products could also be used as a scaffold for reconstruction of skeletal and cartilaginous tissues. Nevertheless, it has not clearly been shown, whether this idea has been validated and which results have been obtained. After the paper I still feel thirsty for knowledge and I would appreciate to extend the paper with data providing this information. Pubmed offers nowadays access to 532 publications on wood derived products for medical applications (i.e. [[1]] and [[2]]) that would help Authors to extend the knowledge of their paper.

Nevertheless, despite the mentioned above remarks I found thew paper very interesting and worth publishing.

Some minor suggestions:

The title: since Authors do not discuss renovation the processes of wood as a material for bone implants, I would suggest to change it into “Wood-derived products as components for bone implants – literature review”.

I am not an expert in English; thus, I would suggest to consult the linguistic aspects of the paper with professional editor. Nevertheless, I think that in line 19th an addition of “is” between words “it” and “also” will improve the structure of the sentence.

[1] Xu W, Wang X, Sandler N, Willför S, Xu C. Three-Dimensional Printing of Wood-Derived Biopolymers: A Review Focused on Biomedical Applications. ACS Sustain Chem Eng. 2018 May 7;6(5):5663-5680. doi: 10.1021/acssuschemeng.7b03924. Epub 2018 Mar 27. PMID: 30271688; PMCID: PMC6156113.

[2] An Overview on Wood Waste Valorization as Biopolymers and Biocomposites: Definition, Classification, Production, Properties and Applications.

Ferrari F, Striani R, Fico D, Alam MM, Greco A, Esposito Corcione C.

Polymers (Basel). 2022 Dec 16;14(24):5519. doi: 10.3390/polym14245519.

PMID: 36559886 Free PMC article. Review.

Author Response

Dear reviewer,
Thank you for Your valuable recommendations for our article. Few points to answer Your propositions:

  1. Regarding mechanical properties of the biomaterials. Among collected literature review only few studies (Andze, Andzs et al. 2022, Qian, Kang et al. 2008, Tampieri, Sprio et al. 2009, Finardi and Sprio 2012) focused on studying mechanical properties of the researched materials. Accordingly, we have reported their results. Other in vitro and in vivo studies did not discuss mechanical properties of the researched materials as they focused on biocompatibility and osteoconductivity. Studies You mentioned focus on extensive wood processing, as a result only cellulose derived polymers are left. In these studies bone surgery is not the main focus and is only slightly mentioned. We have discussed cellulose and bioceramics in sections 2.5 and 2.6. Whereas we are focusing our research on developing implants for orthopedic surgery.
  2. Regarding the title, our main goal in this study was to focus primarily on pure wood as a bone implant (sections 2.1 – 2.5). We gathered all studies we could find on individual species. We added 2.5 and 2.6 sections to expand the article and meet the Journal’s expectations for review articles. Thus, we would like to keep the original title to emphasize our interest in pure wood implants.
  3. The entire text has been reviewed by an English linguistic specialist.